# Improving Anaerobic Digestion of Brewery and Distillery Spent Grains through Aeration across a Silicone Membrane

Zachary P. Berry [1], John H. Loughrin [1,*] , Stuart Burris [2], Eric D. Conte [2], Nanh C. Lovanh [1] and Karamat R. Sistani [1]

[1] Food Animal Environmental Systems Research Unit, Agricultural Research Service, United States Department of Agriculture, 2413 Nashville Road, Suite B5, Bowling Green, KY 42101, USA; zachary.berry2@usda.gov (Z.P.B.); nanh.lovanh@usda.gov (N.C.L.); karamat.sistani@usda.gov (K.R.S.)

[2] Department of Chemistry, Western Kentucky University, 1906 College Heights Boulevard, Bowling Green, KY 42101, USA; stuart.burris@wku.edu (S.B.); eric.conte@wku.edu (E.D.C.)

* Correspondence: john.loughrin@usda.gov; Tel.: +1-270-781-2260

**Abstract:** An increase in the number of independent breweries and distilleries has led to an increase in the amount of spent grains with inadequate means of disposal. One option for disposal is as feedstock for anaerobic digestion if digester stability is ensured. In this study, brewers' spent grain and distillers' spent grain were used as substrate for anaerobic digestion for 32 weeks. The digestate was treated by recirculation through a silicone hose located in an external tank filled with saline solution. The hose served as a permeable membrane allowing for the passage of gases. The recirculation tanks were fitted with check valves to maintain three pressure/gas regimes: 26 mm Hg $N_2$, 26 mm Hg aeration or 100 mm Hg aeration. A fourth digester was operated with no recirculation as the control. These treatments were chosen to determine if differences in digester stability, wastewater treatment efficiency, and biogas production could be detected. A combination of dairy and swine manure was used as seeding to provide a methanogenic consortium and bicarbonate buffering. However, despite trying to provide for adequate initial bicarbonate buffering, all four digesters had low initial buffering and consequently low pH as short-chain fatty acids accumulated. After six weeks, bicarbonate buffering and pH increased as methane production increased, and short-chain fatty acids decreased. Later, despite the fluxes of $O_2$ and $N_2$ across the silicone membrane being very low, differences between the various treatments were noted. The pH of the digestate treated by $N_2$ recirculation was lower than the other digesters and decreased further after distillers' spent grain was substituted for brewers' spent grain. Aeration at a pressure of 26 mm Hg and 100 mg Hg increased biogas production compared to other treatments but only significantly so at 100 mm Hg. These results suggest that partial purging of dissolved gases in anaerobic digestate by the small fluxes of $N_2$ or $O_2$ across a permeable membrane may affect digester performance.

**Keywords:** anaerobic digester; bicarbonate; biogas; brewers' spent grain; carbon dioxide; distillers' spent grain; greenhouse gases; methane

## 1. Introduction

According to the Brewers' Association of United States, there has been an increase in the number of small breweries by approximately six thousand in the past two decades. The primary cause is due to a rise in the number of independent craft breweries. Craft breweries account for over 7100 of the breweries in the United States, with a craft brewery defined as being operated by a small, independent operator (not run by an alcohol industry brewer) with an annual production of fewer than 6 million barrels per annum [1].

With the increase in small, independent breweries comes an increase in waste that requires disposal without the means and infrastructure available to larger facilities. The major components of the waste produced by breweries are brewers' spent grain (BSG),

hops, yeast, and wastewater. According to Mussatto et al., the number one waste product that breweries produce is BSG [2].

In rural areas, an arrangement is often made with local farmers, where, provided BSG meets specific quality standards, it can be used as animal feed. However, before it may be used as such, it must meet quality standards to protect animal health [3]. In urban areas, BSG may be disposed of in various ways such as landfilling, via municipal sewage, and biorecycling. A promising, though underutilized, disposal option for BSG is using it as a feedstock for anaerobic digestion [2,3].

However, BSG is a complex feedstock for anaerobic digestion, with much of the readily fermentable substrates extracted during the malting process. The residuals are largely composed of β-glycans such as cellulose, hemicellulose, and arabinoglycans as well as lignin and other aromatic substances that are slowly degraded in anaerobic environments [2,4]. In addition, it has been suggested that biogas production from BSG is often hampered by a lack of trace elements such as Cu, Mo, and Zn that act as cofactors in fermentation as well as methanogenesis and that a lack of these elements often leads to digester instability and failure [5–8].

The perceived lack of essential nutrients in anaerobic digestion has been addressed by mineral supplementation [6] and/or co-digestion with animal manures. Manures may contain many trace elements [9], so that presumed deficiencies of BSG in this regard could possibly be addressed by co-digestion with manure.

BSG, although depleted of much of its readily fermentable substate during the malting process, is capable of copious short-chain fatty acid (SCFA) production during fermentation including lactic, acetic, propanoic, butyric and iso-butyric acids [10], with the SCFA composition depending on species of bacteria present. SCFA production may overpower any buffering present in digestate, especially during digester startup when rates of methanogenesis are low. In contrast, animal wastes are usually highly buffered [11], especially by bicarbonate ($HCO_3^-$), so that much of the positive effect of co-digestion with animal wastes that is attributed to mineral supplementation could possibly be attributed to enhanced buffering preventing the acidification of digestate. Methanogenesis is sensitive to pH with the optimal range reported as 6.7–7.4 [12]. Therefore, whether due to enhanced buffering or mineral supplementation, the addition of animal manure to BSG digestate is likely to enhance digester stability during digester startup when buffering is low, and minerals may be lacking due to the relatively intact nature of the grains following alcoholic fermentation.

In the present study, BSG was investigated as the primary substrate for anaerobic digestion. Proper anaerobic digester startup is critical to ensuring stable biogas production and organic matter mineralization [13]. Because maintaining adequate buffering is critical to anaerobic digester success, not only in the early stages when volatile fatty acids accumulate due to limited methanogenesis, but also in later stages of digestion, the digesters were seeded with a mixture of animal manure to ensure some level of $HCO_3^-$ buffering and more quickly build populations of methanogenic Archaea [5].

In previous research, it was shown that recirculating swine waste in an anaerobic digester through a silicone membrane located in an external aerated chamber could greatly enhance $HCO_3^-$ buffering and boost biogas quality [14,15]. Presumably, this occurred by aeration of the waste and the subsequently mineralized carbon was sequestered as $HCO_3^-$.

This limited carbon dioxide concentration in the gaseous phase and furthermore limited carbon dioxide in the biogas by raising digestate pH and sequestering the $CO_2$ as $HCO_3^-$. In that BSG is capable of considerable SCFA production, there is a danger that acidification of the digestate could occur that threatens the growth of methanogens and causes digester upset [16]. Therefore, in addition to co-digestion of BSG with animal manure, enhancing $HCO_3^-$ buffering by transmembrane aeration could also help ameliorate the effects of digestate acidification.

Consequently, for this experiment, anaerobic digestion of BSG was studied under four regimens: recirculating digestate through a silicone membrane under two pressures of aeration, recirculation of the wastewater through a silicone membrane in an anaerobic

environment (26 mm Hg N$_2$), and with no recirculation. The various treatments were compared and contrasted in regard to biogas production and its quality, pH and bicarbonate buffering as well as other wastewater quality parameters. In addition, the digesters were seeded with animal manure, and the manures and spent grains feedstocks were analyzed for metallic cofactor concentrations. In this manner, we hoped to determine if seeding with manure and transmembrane aeration via recirculation through a silicone hose could enhance process stability and biogas production.

## 2. Materials and Methods

### 2.1. Digester Descriptions

The digesters were kept in a greenhouse at 26.7 °C, with a range of approximately 22.0–31.1 °C and using a natural light cycle. The trial of the digesters was performed from 1 May to 18 December 2018. Primary anaerobic digesters were constructed from 208 L (55 gallon) closed-top plastic barrels with a height of 87 cm and a diameter of 59 cm (Uline Inc., Pleasant Prairie, WI, USA). The barrels were painted dark green to limit light penetration. The barrels were fitted with 5.1 cm-diameter PVC (polyvinylchloride) pipes with manual ball valves near their bottom that served as drains and ports for the withdrawal of sludge samples. The barrels also had two 5.1 cm-diameter threaded openings in the top. One of these openings was used as the waste inlet and accommodated a fitting of a 5.1 cm-diameter PVC pipe with a manual valve. A reducing union inside the tank led to a 3.8 cm-diameter PVC pipe, the outlet of which was below the digestate surface in order to avoid venting of the digester during feeding. The other 5.1 cm opening in the top of the digester was used to house a 1.27 cm-diameter PVC pipe on which a 10 W, 120 V, 0.5 A float switch (Anndason Industrial, amazon.com, Seattle, WA, USA, accessed on 25 January 2022) was installed to maintain a digestate volume of 150 L. The float switch controlled a 120 V electrically actuated, stainless steel, normally closed ball valve with an aperture of 2.54 cm (BACOENG, Suzhou Jianli Machinery and Equipment Co., Ltd., Suzhou, China) connected to a 2.54 cm-diameter PVC pipe that served as the waste outlet.

The waste outlet was connected by a 38 cm length of PVC pipe to a 56 L (15 gallon) secondary aerobic digester (ULINE Inc.). The secondary digesters were fitted 3 m lengths of 6.35 mm porous irrigation tubing (NDS Inc., Lindsay, CA, USA) placed at the bottom and which was sealed at one end with silicone caulk. The soaker irrigation tubing was supplied with air via an air compressor at 200 mL min$^{-1}$ for 30 min four times daily. The digestate volume in the secondary digester was maintained at 45 L using a float switch and electrically actuated ball valve as in the primary anaerobic digester. Each secondary aerobic digester had 100 bio balls that served as support media for microbial growth (thepondguys.com, Armada, MI, USA, accessed on 25 January 2022).

The outlet of the secondary digester led to a 190 L plastic water trough that served as a wastewater lagoon (Tractor Supply Co., Brentwood, TN, USA). This trough was used as a source of water with which to mix feed to supply the primary digester.

Each primary and secondary digester had a 6.35 mm hole drilled into the top to accommodate a Luer fitting and a gas sampling port consisting of a three-way Luer lock fitting. One arm of the fitting was used to take gas samples by means of a syringe and, on the primary anaerobic digester, the other was fitted with 6.35 mm tubing and led to a wet tip flowmeter (wettipgasmeter.com, accessed on 25 January 2022). The side of both the primary anaerobic and secondary aerobic digester had another 0.635 cm-diameter port with a Luer fitting and two-way valve for liquid analyses at the height of 45 cm on the primary digester and 38 cm on the secondary digester. All pipe and tubing connections to the tanks were made with Uniseal pipe to tank fittings (US Plastic Inc., Lima, OH, USA). An illustration of a digester system equipped with wastewater recirculation in a supplemental tank is given as Figure 1.

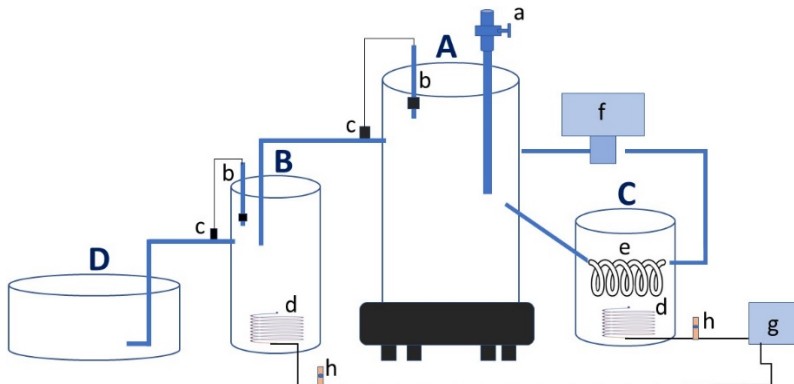

**Figure 1.** Schematic representation of digester system. A. Primary digester tank. B. Secondary aeration tank. C. Aeration recirculation tank. D. Water trough for mixing of feed. a. Feed inlet with manual ball valve. b. 10 W, 120 V AC float level switches. c. Electrically actuated 120 V AC ball valves. d. Aeration hose. e. Silicone tubing. f. Peristaltic pump. g. Aeration pump. h. Flowmeter.

### 2.2. Digestate Recirculation

One primary digester had no provision for wastewater recirculation into an auxiliary tank and served as a control. The others had additional fittings placed 47 cm from the bottom of the digesters that fed the inlet of the recirculation tank and 58 cm from the bottom that served as the inlet back into the primary digester. PVC tubing was used to transfer digestate to the recirculation tank and back to the primary digester. Inside the recirculation tank was a 3 m length of 16.7 mm-outer-diameter nylon braid-reinforced silicone tubing with a wall thickness of 3.6 mm and inner diameter of 9.5 mm (US Plastic Inc., Lima, OH, USA). The recirculation tank had a volume of 45 L (Greif Industrial, Carol Stream, IL, USA) and was filled with 37 L of a 5% synthetic seawater solution (Instant Ocean, Mentor, OH, USA). Wastewater was recirculated through the silicone hose at a rate of 100 mL min$^{-1}$ by means of a peristaltic pump and a short length of Pharmed tubing (US Plastic Inc.) which represented a nominal recirculation of the digestate once every 1500 min or slightly over one day. Each recirculation tank had 3 m lengths of 6.35 mm porous irrigation tubing at the bottom of the tank which was sealed at one end with silicone caulk. The tubing was supplied with air via a compressor or N$_2$ via a gas cylinder at a rate of 100 mL min$^{-1}$ controlled by 15 cm rotameters. The internal pressure of the aeration tank was maintained at 26 mm Hg for one aerated tank and the N$_2$ treatment tank at 100 mm Hg for the second aerated tank.

### 2.3. Digester Operation

Initial feedstock for the digesters was a mixture of brewers' spent grain and water from the lagoon. Initially, the digesters were fed 1.0 kg of brewers' spent grain and 6.0 L lagoon water. As the experiment went on, the feed stock was changed to 2 L of distillers' spent grain (DSG), 4 L of lagoon water and 2 kg of BSG. Fifteen grams of BSG was suspended in deionized water, stirred and the resultant solution had a pH of 5.93. The BSG had a moisture content of 77.2% and a volatile solids (VS) content of 11.7%. The DSG was had a pH of 3.4 and a vs. content of 21%. All feedstocks were stored at 4 °C prior to use.

The digesters were filled with sufficient tap water to fill both the primary and secondary stages and ensure proper outlet valve operation. On the first day of operation, 90 L of wastewater from a facultative lagoon serving a swine farrowing operation was added to the system, followed the next day by 2.0 kg of waste obtained from a local dairy. The following week, the digesters were fed 1.0 kg BSG mixed with 10 L system effluent from the plastic trough giving a hydraulic retention time of 105 days for the primary digester and 31.5 days for the secondary aeration tank. This feeding schedule was maintained for 15 weeks when the brewery which supplied BSG closed, whereupon 2.0 L distillers' spent grain (DSG) suspended in 10 L system effluent was substituted. Another source of BSG

was found, and from week 23 onwards, the digesters were fed 1.0 kg BSG and 2.0 L DSG suspended in 10 L of effluent until the end of the experiment. Due to a decline in pH and $HCO_3^-$ buffering, each digester was fed a mixture of 500 g each of $Ca(OH)_2$ and $CaCO_3$ on week 14.

*2.4. Analyses*

All samples for analyses were collected prior to the once-weekly feeding. Dissolved gases were analyzed using gas chromatography (GC) as described [17]. A volume of 0.5 mL of water was withdrawn from the wastewater sampling port on the side of the digesters and injected using an 18-gauge needle through the septum on a 20 mL headspace vial containing 9.5 mL 0.1 N HCl. Samples were analyzed on a Varian CP-3800 GC (Agilent Technologies, Santa Clara, CA, USA) modified for greenhouse gas analysis by The RSC Group (Katy, TX, USA). A volume of 1 mL of headspace sample was injected by syringe and split by valve switching onto separate 1.8 m $\times$ 1.6 cm columns packed with 80/100 mesh Hay Sep Q. Carbon dioxide was detected by a thermal conductivity detector (TCD) operated at 120 °C and CH4 was detected with a flame ionization detector (FID) operated at 250 °C. Concentrations of dissolved $CO_2$ ($sCO_2$), $HCO_3^-$, and dissolved $CH_4$ ($sCH_4$) were calculated using dimensionless Henry's constants and adaptation of the Henderson–Hasselbalch equation [17,18].

Biogas concentrations of $CO_2$ and $CH_4$ were measured using an Agilent model 7890b GC also modified by the RSC Group. A volume of 2 mL of the headspace gas was injected by syringe and split by valves switching the gas onto five different columns. Carbon dioxide was detected by a TCD operated at 120 °C and $CH_4$ was detected with an FID operated at 250 °C.

Metals were determined by inductively coupled plasma spectroscopy-optical emission spectroscopy (ICP-OES), using EPA Methods [19]. Duplicate 10 mL sludge samples were digested for 45 min. at room temperature in a Teflon microwave digestion vessel after the addition of 9 mL of ACS reagent-grade $HNO_3$ and 3 mL HCl. The samples were then digested for 12 min at 175 °C in a microwave (Mars 6 microwave oven, CEM Corp., Matthews, NC, USA). The samples were cooled to room temperature and filtered through a Whatman no. 42 filter prior to analysis on an ICP-OES (ICP-OES 5110, Agilent Technologies).

Short-chain fatty acids were analyzed by high-performance liquid chromatography (HPLC, Thermo Fisher Scientific, Waltham, MA, USA) after filtration of digestate through a 0.2 μm-pore-size nylon filter (Sigma-Aldrich Corp., St. Louis, MO, USA). Samples were analyzed using a RHX monosaccharide column (Phenomenex, Torrance, CA, USA). Sulfuric acid (5 mM) was used as the isocratic mobile phase and at a flow rate of 5 mL min$^{-1}$ at a temperature of 65 °C. Detection was performed with a photodiode array detector at a wavelength of 210 nm.

Data were examined by analysis of variance using PROC ANOVA in the Statistical Analysis System for Windows (SAS) version 9.4 (SAS Institute, Cary, NC, USA). Means were contrasted by a Studentized Tukey's test and significant differences were determined at $p < 0.05$.

## 3. Results

*3.1. Silicone Membrane Properties*

Previously, it was shown that recirculating swine waste from an anaerobic digester through a silicone membrane located in an externally aerated chamber could greatly enhance $HCO_3^-$ buffering and boost biogas quality [14,15]. As discussed below, the results of the present study differed from those of the previous study. The reasons for this are related to differences in the experimental setup and in particular in the way the aeration of the digestate was accomplished.

In contrast to Loughrin et al. [14,15], the constant pressure in the recirculation tank (26 mm Hg or 100 mm Hg) was higher than that of the anaerobic digester, which had an estimated pressure of 6–9 mm Hg. In the earlier experiment, the aeration tank was not

pressurized, but allowed to vent with no backpressure. The daily pressure in the anaerobic digesters varied greatly, however, from 0 mm Hg above atmospheric pressure, after gas production was measured by venting the digester, up to approximately 150 mm Hg. The pressure gradient across the silicone membrane therefore favored flux of gas from the anaerobic digester to the aeration tank except for brief periods when the anaerobic digester was vented to measure daily biogas production.

It was surmised that increases in $HCO_3^-$ buffering were due to the low-level microaeration of the wastewater with the resultant $CO_2$ being converted to $HCO_3^-$ due to the relatively high pH of the digestate [14,15]. Loss of $CH_4$ through the silicone hose would account for the drop in biogas production. Given the magnitude, and direction of, the pressure gradient across the silicone membrane, however, this interpretation was false, and little or no $O_2$ flux from the aeration tank to the anaerobic digester occurred. On the other hand, this arrangement did favor loss of $CH_4$ from the anaerobic digester to the aeration tank. The increase in $HCO_3^-$ buffering that was observed would likely be the result of $CO_2$ loss through the membrane shifting the equilibrium from water-solvated $CO_2$ to $HCO_3^-$:

$$CO_2(s) + H_2O \rightleftharpoons H_2CO_3^- \rightleftharpoons H^+ + HCO_3^- \tag{1}$$

In the present research, the membrane was a 3 m length of silicone hose with a 0.95 cm diameter that was reinforced with a nylon braid and a wall thickness of 3.6 mm. The internal (anaerobic) membrane area was 898 $cm^2$ and the external (aerated) membrane had an area of 1590 $cm^2$. In the previous experiment, these values were 761 and 1050 $cm^2$, respectively [14,15]. With digestate volumes of 37.9 and 150 L in the former and present study, respectively, it can be seen that the effective ratio of the membrane area to digestate volume was much higher in the previous research.

Methyl silicone rubber (polydimethylsiloxane, PDMS) is a unique polymer with the highest gas permeability of any known elastomer [20]. Permeability is also affected by solubility and molecular mass of a given gas, pressure differential across the membrane, temperature, and membrane thickness. PDMS was used as the membrane both in previous research [14,15] and here. Other than flux direction, area of the membrane and thickness, the main difference between the experiments was that the tubing used in the present experiment had nylon braiding reinforcement. Due to its limited area, assumptions were made that the nylon braiding did not significantly affect gas permeability. The chief gases of concern in the present experiment are $O_2$ and $N_2$. Carbon dioxide was not considered given the direction of gas flux and due to atmospheric $CO_2$ concentrations being negligible compared to those of the anaerobic digester.

Gas permeates through the silicon tubing due to the pressure differential across the tubing of either 26 or 100 mm Hg. The rate at which it permeates is dependent on the pressure differential across the membrane, the thickness of the membrane and the membrane's surface area. It is described by Equation (2):

$$Permeability\ [Barrer] = (q \times t)/(A \times s \times \Delta p) \tag{2}$$

where $q$ is the flux of gas through the membrane at $STP_{(cm3)}$ per time $s$, $t$ is the thickness of the membrane in cm, $A$ is the area of the membrane in $cm^2$, and $\Delta p$ is the pressure differential across the membrane in cmHg [21]. $P$, the silicone permeability coefficient is expressed in Barrers ($10^{-10}\ cm^3\ _{(STP)} \cdot cm/cm^2 \cdot s \cdot cmHg$) and is equal to 280 Barrers for $N_2$ and 600 Barrers for $O_2$ [22].

Assuming a partial pressure of 59.3 cm Hg for $N_2$ in the atmosphere gives a $N_2$ pressure differential across the silicone membrane of 61.3 cm Hg at 26 mm Hg aeration and 67.1 cm Hg at 100 mm Hg aeration. For the 26 mm $N_2$ recirculation treatment, the pressure differential across the membrane was 78.6 cm Hg. Assuming a partial pressure of 16.0 cm Hg for $O_2$ in the atmosphere gives a pressure differential across the silicone membrane of 16.5 cm Hg for $O_2$ at 26 mm Hg aeration and 18.1 cm Hg for $O_2$ at 100 mm Hg aeration.

Using these pressure differentials across the silicone tubing gives a calculated $N_2$ flux of 0.9 and 3.5 mL $h^{-1}$ for 26 and 100 mmHg aeration, and for $O_2$, 0.49 and 1.9 mL $h^{-1}$ at pressure differentials of 26 and 100 mmHg, respectively. For a pressure differential of 26 mm $N_2$, the flux was calculated as 1.2 mL $h^{-1}$.

Although these fluxes were quite small, noticeable differences in wastewater chemistry and biogas quality were noted as described below. Though the mechanism underlying these effects is unknown, it may have been due to low-intensity, long-term, sparging of $CH_4$ and $CO_2$ from the digestate in the three recirculation treatments. Thus, lower concentrations of soluble gases in the digestate could lead to greater biogas production. This would be coupled with the effects of, and low-intensity, long-term, microaeration in the two aerated recirculation treatments.

### 3.2. Biogas Production and Wastewater Analysis

The digesters were initially fed 1.0 kg BSG along with 2.0 kg cattle manure and 90 L swine lagoon waste. The pH of the BSG was 3.49, whereas the pH of the cattle and swine waste was 6.5 and 7.8, respectively. The addition of animal waste helped ensure higher initial digester pH and seeding with methanogenic consortia to assist with rapid digester startup. Animal wastes typically have high levels of $HCO_3^-$ buffering [11], largely due to the activity of carbonic anhydrases coupled with Cl-$HCO_3^-$ and VFA-$HCO_3^-$ antiporters in the colon [23]. Due to the activity of these, $HCO_3^-$ concentration increases in feces when exchanged for chloride and nutrients in the form of VFA, particularly in ruminants where VFA represents the majority of caloric uptake [24].

Therefore, the anaerobic digester startup of animal waste is usually more stable than the digestion of other wastes that lack inherent bicarbonate buffering. Fresh BSG usually has a pH of 4.5 or below and produces considerable SCFA upon fermentation [2]. Fresh DSG is also an easily fermentable substrate and may have a pH below 3.5 [25]. High initial $HCO_3^-$ buffering is necessary, therefore, to ensure stable anaerobic digestion of both feeds.

Despite seeding of the digesters with combined swine and dairy waste, the initial bicarbonate buffering of all four digesters was low, averaging approximately 5 mM, and declined to an average of 3.3 through week 6 (Figure 2). It rapidly increased thereafter and averaged 16.5 mM through week 16. Despite this, the pH of the digesters, after fluctuating between approximately 6.3 and 6.9 during the first six weeks, rose considerably and remained fairly stable at a pH of approximately 7.0. The digester employing 26 mm Hg $N_2$ recirculation was the exception to this, showing a decline in pH. At week 15, bicarbonate buffering also declined, and both pH and $HCO_3^-$ concentrations remained lower in the $N_2$ recirculation than the other three digesters for the rest of the experiment. As a consequence, $sCO_2$ was also highest in this treatment (Figure 2).

Bicarbonate buffering decreased in all four treatments in response to the feeding of DSG from weeks 15 through week 23. The pH of the digestate recovered when feeding of BSG recommenced on week 22 with the exception of the 26 mm Hg $N_2$ treatment. In this treatment, pH and $HCO_3^-$ concentrations apparently declined too much to allow for the recovery of microbial populations and as a consequence, gas production from the digester coupled with $N_2$ recirculation averaged only approximately 60% that of the other three digesters from week 25 onwards. In contrast to previous research [14,15], $HCO_3^-$ buffering was not enhanced relative to the control treatment, so $sCO_2$ concentrations were not reduced, and consequently biogas $CO_2$ concentration was not decreased. As stated in Section 3.1, this is likely due to retention of $sCO_2$ in the digester due to net gas influx from the aerated recirculation tank into the digester increasing carbonic acid ($H_2CO_3$) concentration. While $H_2CO_3$ rapidly dissociates into $HCO_3^-$ and $H_3O^+$ at higher pHs, this would not so much be the case for the $N_2$ recirculation treatment as for the control and aeration-treated digesters.

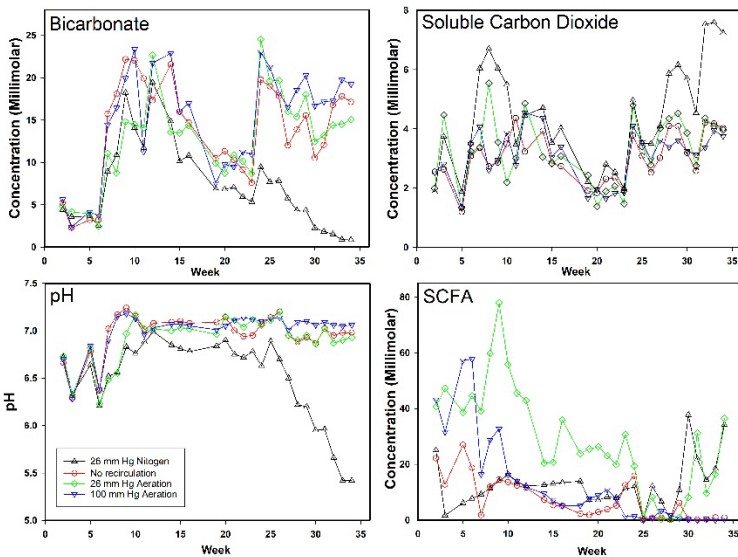

**Figure 2.** Selected buffering components, bicarbonate, soluble carbon dioxide, pH and short-chain fatty acids of the anaerobic digestate of the four treatments.

Weekly biogas production from the 26 mm Hg aeration and 100 mm Hg averaged approximately 5% and 27% higher than that of the control digester (Table 1). Mostly due to its low pH, biogas production from the $N_2$ recirculation treatment was much lower than that of the control. Weekly $CH_4$ production of the $N_2$ recirculation, 26 mm Hg, and 100 mm Hg aeration treatments was 42, 100, and 126 percent that of the control treatment.

**Table 1.** Biogas and digestate characteristics of primary anaerobic digesters *.

| | Recirculation Treatment | | | |
|---|---|---|---|---|
| **Parameter** | **None** | **26 mm Nitrogen** | **26 mm Air** | **100 mm Air** |
| | Biogas characteristics | | | |
| Weekly biogas (L) | $78.4 \pm 11.1$ b | $42.3 \pm 4.79$ c | $82.7 \pm 9.01$ b | $99.9 \pm 9.69$ a |
| Methane (mg $L^{-1}$) | $514 \pm 23.6$ a | $382 \pm 30.9$ b | $473 \pm 27.2$ a | $499 \pm 24.1$ a |
| Carbon dioxide (mg $L^{-1}$) | $382 \pm 14.0$ b | $509 \pm 29.6$ a | $406 \pm 21.0$ b | $366 \pm 13.1$ b |
| Methane (g $week^{-1}$) | $44.3 \pm 6.74$ b | $18.7 \pm 2.54$ c | $44.4 \pm 5.28$ b | $55.7 \pm 5.79$ a |
| Carbon dioxide (g $week^{-1}$) | $30.0 \pm 4.10$ b | $22.6 \pm 2.80$ c | $32.3 \pm 3.73$ a,b | $36.7 \pm 3.82$ a |
| | Digestate characteristics | | | |
| pH | $6.98 \pm 0.04$ a | $6.50 \pm 0.08$ b | $6.92 \pm 0.04$ a | $7.01 \pm 0.04$ a |
| Bicarbonate (millimolar) | $14.2 \pm 1.06$ a | $7.42 \pm 0.95$ b | $13.2 \pm 0.98$ a | $15.4 \pm 1.14$ a |
| Solvated carbon dioxide (millimolar) | $3.05 \pm 0.16$ b | $4.49 \pm 0.33$ a | $3.31 \pm 0.21$ b | $3.10 \pm 0.17$ b |
| Solvated methane (millimolar) | $18.0 \pm 3.37$ a | $10.2 \pm 1.83$ c | $15.4 \pm 2.76$ b | $18.5 \pm 3.43$ a |
| Ammonium (mg $L^{-1}$) | $195 \pm 21.5$ ab | $197 \pm 23.2$ ab | $177 \pm 23.1$ b | $208 \pm 22.5$ a |
| Nitrate (mg $L^{-1}$) | $1.31 \pm 0.33$ a | $1.59 \pm 0.37$ a | $1.09 \pm 0.28$ a | $1.13 \pm 0.33$ a |
| Nitrite (mg $L^{-1}$) | $3.07 \pm 1.31$ a | $2.60 \pm 0.99$ a | $3.21 \pm 1.27$ a | $3.19 \pm 1.19$ a |
| Phosphate (mg $L^{-1}$) | $10.4 \pm 2.38$ a | $9.78 \pm 2.38$ a | $12.7 \pm 3.18$ a | $13.4 \pm 3.52$ a |
| Sulfate (mg $L^{-1}$) | $0.49 \pm 0.13$ a | $0.97 \pm 0.38$ a | $0.42 \pm 0.08$ a | $0.43 \pm 0.10$ a |
| Chemical oxygen demand (mg $L^{-1}$) | $2510 \pm 291$ c | $6700 \pm 616$ a | $3700 \pm 357$ b | $2320 \pm 298$ c |

* Data represent the mean $\pm$ standard error of the mean of 28 weekly determinations. Within a row, means labelled by the same letter are not significantly different by analysis of variance with Tukey's multiple comparisons test at $p < 0.05$.

Due to the lower pH of its digestate, $sCO_2$ concentrations were significantly higher for the treatment involving recirculation through the $N_2$-treated membrane and consequently, $HCO_3^-$ buffering was significantly lower than in the other treatments. Solvated $CH_4$ concentrations were lower in the $N_2$ recirculation treatment, which resulted in low $CH_4$ production.

Biochemical methane potential (BMP) is a means of testing the degradability of feedstocks in anaerobic digestion [26]. Theoretically, BMP reports a maximum potential yield of 0.35 L of $CH_4$ per g of VS. BMP tests are usually run as batch tests to assess the degradability of feedstocks and resultant biogas yields, however; and in continuously fed digesters, may give variable and misleading results as previously fed feedstock is accumulated and later digested. In this experiment, $CH_4$ yields during week 11 when the digesters were fed 1 kg BSG per week (117 g VS) and the digesters were producing over 500,000 ppm $CH_4$, $CH_4$ yields were 0.37, 0.57, 0.78, and 1.1 L $CH_4$ per g vs. for the control, $N_2$ recirculation and 26 and 100 mm aeration recirculation treatments, respectively.

During week 22 when the digesters were fed 420 g vs. per week of DSG, apparent $CH_4$ yields declined to 0.14, 0.12, 0.18, and 0.25 L $CH_4$ per g vs. for the control, $N_2$ recirculation and 26 and 100 mm aeration recirculation treatments, respectively. By week 32, when the digesters were fed 537 g vs. from a combination BSG and DSG, $CH_4$ yields were 0.37, 0.15, 0.32, and 0.41 L $CH_4$ per g vs. for the control, $N_2$ recirculation and 26 and 100 mm aeration recirculation treatments, respectively. Beyond emphasizing the enhanced $CH_4$ yields from the 100 mg aeration recirculation, these results also emphasize the limitation of applying BMP assays to continuously fed anaerobic digesters.

No variation in the concentrations of $NO_2^-$, $NO_3^-$, $PO_4^{3-}$, or $SO_4^{2-}$ due to treatment were noted. In the case of $NO_3^-$ and $NO_2^-$, this likely indicates that the amount of aeration was insufficient in any treatment to support nitrification/denitrification. A low but steady concentration of $SO_4^{2-}$ in all four digesters likely indicates little activity of sulfate reducing bacteria or archaea. Inorganic $PO_4^{3-}$ was higher in the aerated treatments, but this difference was insignificant. Still, this could possibly be due to a greater degree of feedstock breakdown resulting in release of inorganic $PO_4^{2-}$ from organically bound forms, especially for the digestate recirculated through the membrane with 100 mm Hg aeration since this treatment produced significantly more biogas and $CH_4$ than the other treatments.

The concentrations of SCFA found in the primary anaerobic digestate are presented in Table 2. Concentrations of SCFA were high in all four treatments in the first weeks after digester startup, with the most predominant acids being formic, acetic, and propanoic. Specific SCFA are characteristic of bacterial species that produce them, although acetic acid may be produced by pathways other than fermentation [27].

**Table 2.** Average short-chain fatty acid concentration in primary anaerobic digesters *.

| | | Recirculation Treatment | | |
|---|---|---|---|---|
| **Acid** | **None** | **26 mm Nitrogen** | **26 mm Air** | **100 mm Air** |
| | | **Concentration (Micromolar)** | | |
| Lactic | 54.4 ± 16.1 a | 50.1 ± 16.8 a | 97.6 ± 28.9 a | 174 ± 34.0 a |
| Formic | 2490 ± 517 b | 6700 ± 831 a | 3300 ± 1270 b | 696 ± 166 b |
| Acetic | 1600 ± 471 b | 1950 ± 592 b | 16,500 ± 2280 a | 5720 ± 1870 b |
| Propanoic | 2600 ± 634 b | 2690 ± 435 b | 6070 ± 1090 a | 3580 ± 985 b |
| 2-Methylpropanoic | 464 ± 158 a | 991 ± 249 a | 645 ± 155 a | 1570 ± 1000 a |
| Butyric | 91.6 ± 41.1 b | 346 ± 81.2 b | 1750 ± 557 a | 1070 ± 380 ab |
| Total identified | 7300 ± 1350 b | 12,700 ± 1530 b | 28,400 ± 3480 a | 12,700 ± 3010 b |

* Data represent the mean ± standard error of the mean of 28 weekly determinations. Within a row, means labelled by the same letter are not significantly different by analysis of variance with Tukey's multiple comparisons test at $p < 0.05$.

Typically, lactic acid-producing bacteria are aerotolerant Gram-positive bacteria belonging to the order Lactobacillales, although bacteria in other orders such as Bifidobacteriales and Bacillales are also capable of lactic acid production. They may be classified biochemically as homolactic fermenters, which produce lactic acid as a sole product or heterolactic fermenters which produce lactate as well as $CO_2$ and either ethanol or acetate. In addition, facultative heterolactic acid fermenters utilize either pathway depending on the fermentable substrates available [28].

Acetic acid is a common fermentation end product and the ability to produce it as a fermentation end product is widespread in anaerobic bacteria and acetogens which are anaerobic bacteria or Archaea that produce acetyl Co-A and hence acetate through the reduction of $CO_2$ by $H_2$ [29]. Additionally interesting is acetate production via fermentation by acetic acid bacteria. These are obligate aerobes belonging to the family Acetobacteraceae [30,31]. These bacteria are important in the production of certain fermented foods such as cocoa as well as spoilage of wine and beer where they may form a thin film on the beverage surface. In the context of the present study, it is interesting to note that the aerated treatments had the highest concentrations of acetic acid.

The two aerated treatments also had the highest concentrations of butyric acid. The most well-known butyrate fermenters are obligate anaerobes belonging to the order Clostridiales. The relevance of this discussion is that variance in the concentrations of SCFA noted between the treatments is likely related to variation in microbial communities between the four treatments. Variation in the rate of BSG/DSG degradation among the four treatments will also affect the concentrations of SCFA as well as the ability to convert SCFA to acetate and hence $CH_4$; however, insight into how these treatments affect microbial populations will await. It is interesting to note, however, that in Loughrin et al. [14,15], swine waste digestate recirculated through a silicone membrane in an aerated external tank had more clones belonging to the Proteobacteria than did control swine waste digestate. This result indicates that low-level aeration as employed here can potentially cause shifts in bacterial populations in anaerobic digestion and perhaps also in the chemical composition of the digestate.

*3.3. Feedstock ICP*

ICP analysis of the feedstocks used in the experiment is presented in Table 3. The highest concentrations of metals were found in BSG due to its separation into distinct supernatant and grain layers, the latter of which was used as feedstock, whereas DSG formed more of a suspension of grain and residual distillate typically referred to as spent lees. Similarly, concentrations of metals were much higher in dairy manure than in swine lagoon waste due to its higher solids concentration.

**Table 3.** Selected metals found in digester feedstocks.

| | Feedstock | | | |
|---|---|---|---|---|
| **Element** | **Brewers' Spent Grain** | **Distillers' Spent Grain** | **Swine Lagoon Sludge** | **Dairy Manure** |
| | **Concentration (Milligrams $L^{-1}$)** | | | |
| Calcium | 3240 | 73.4 | 129 | 1790 |
| Magnesium | 2890 | 182 | 37.9 | 485 |
| Potassium | 1620 | 560 | 655 | 2724 |
| Sodium | 186 | 11.7 | 1.27 | 10.8 |
| Iron | 210 | 5.14 | 10.4 | 84.1 |
| Zinc | 88.7 | 4.26 | 4.42 | 7.92 |
| Manganese | 52.1 | 4.83 | 1.27 | 10.8 |
| Copper | 31.4 | 0.147 | 2.17 | 2.33 |
| Nickel | 0.90 | 0.02 | 0.037 | 0.18 |
| Molybdenum | bdl * | bdl | bdl | bdl |
| Cobalt | bdl | bdl | bdl | bdl |

* Occurred in concentrations below the detection limit. Data represent the average of two determinations.

Calcium and Mg were the most concentrated metals in all feedstocks except dairy manure in which potassium occurred at the highest concentration, and all four feedstocks also had considerable amounts of Fe, Zn, and Mn. Other metals such as Ni and Co that are reported as required cofactors in methanogenesis [32,33] were present near or below the limit of detection.

Determining optimal metal requirements for the microbial communities involved in anaerobic digestion is difficult due to several factors. Some required transition metals, such as Cu, Ni and Co, apparently exhibit toxicity to microbial communities above certain concentrations [33,34]. Some metals may be present in seemingly adequate concentrations but be poorly bioavailable [35]. For instance, sulfate-reducing bacteria (SRB) produce sulfide, which reacts with metals in wastewater and other contaminated waters to form insoluble sulfide precipitates [36,37]. Confounding this further, many bacterial species can respond to low concentrations of essential metals by excreting ligands, referred to as metallophores, that solubilize metals from such forms as insoluble phosphates and sulfides [38]. Even SCFA may increase the solubility of metals and enhance their bioavailability [35]. Since the chemistry of trace metal chemistry in anaerobic digestion is so complex, methods such as ICP alone are probably inadequate to determine bioavailability of metals in wastewater.

The addition of Mg, K, and Mo to anaerobic digestates of BSG can enhance stability and biogas production [7]. In our study, all digesters showed roughly similar performance through week 25 in terms of pH and biogas production. The digester using 26 mm Hg $N_2$ recirculation, however, had lower $HCO_3^-$ buffering than did the other treatments. After week 25, this treatment had higher $sCO_2$, and consequently lower pH as $HCO_3^-$ buffering continued to decline. As seen in Table 2, all four treatments had considerable SCFA, which may have acted to help solubilize trace amounts of metals [35]. Seeding of the digesters with animal manures at the beginning of the experiment was done with the intention of improving process stability by providing $HCO_3^-$ buffering. The addition of required metallic elements may have been an additional benefit. Regardless, with the exception of the $N_2$ recirculation treatment, the digesters had good process stability as well as good biogas production for approximately eight months after the addition of the manures. The most obvious factor contributing to this stability seems to be ensuring adequate $HCO_3^-$ buffering.

*3.4. Secondary Aerobic Digester Performance*

Chemical oxygen demand in the secondary aeration tanks averaged approximately 52, 35, 48, and 60 percent that of the anaerobic digesters for the control, 26 mm Hg $N_2$, 26 mm Hg air, and 100 mm Hg air treatments, respectively (Table 4). Because the digester treated with recirculation within the $N_2$ recirculation tank had the highest levels of COD (Table 1), the aeration on this treatment was much more effective at removing COD than were the other treatments. It is likely that much of the COD in the digester subjected to the $N_2$ recirculation treatment consisted of relatively small, easily biodegradable substances that were not utilized for methane production, as biogas production was lowest in this treatment (Table 1).

While the pH of the secondary aeration tanks was higher than that of the anaerobic digesters, $HCO_3^-$ concentrations in the aeration tanks and anaerobic digesters were quite similar. The exception to this was for the $N_2$ recirculation treatment due to its low primary digester pH. This treatment consequently also had the greatest increase in $HCO_3^-$ buffering in the aeration tank relative to the digester tank. Due to the increase in pH in the aeration stage, $sCO_2$ was low in all four treatments.

Surprisingly, solvated and other forms of aqueous-phase $CH_4$ (e.g., bubbles attached to suspended matter) were higher in the aeration tanks for all four treatments. This is likely due to both anaerobic niches within the aeration tanks and the carryover of wastewater methane into the aeration tanks. It is likely also an indication of negligible methane oxidation occurring within the aeration tanks.

Similar to the situation in the primary anaerobic digesters, no significant differences were seen between recirculation treatment and concentrations of $NH_4^+$, $NO_3^-$, $NO_2^-$, or $PO_4^{3-}$.

**Table 4.** Wastewater characteristics of secondary aerated digesters.

| Parameter | Primary Digester Treatment | | | |
|---|---|---|---|---|
| | None | 26 mm Nitrogen | 26 mm Air | 100 mm Air |
| Chemical oxygen demand (mg $L^{-1}$) * | $1300 \pm 175$ b | $2350 \pm 153$ a | $1770 \pm 268$ ab | $1390 \pm 138$ b |
| pH [†] | $7.95 \pm 0.04$ a | $7.84 \pm 0.04$ ab | $7.75 \pm 0.04$ b | $7.68 \pm 0.04$ b |
| Bicarbonate (millimolar) [†] | $15.1 \pm 0.98$ a | $15.8 \pm 1.34$ a | $19.3 \pm 1.47$ a | $16.7 \pm 1.23$ a |
| Solvated carbon dioxide (millimolar) [†] | $0.72 \pm 0.14$ a | $0.67 \pm 0.20$ a | $0.98 \pm 0.14$ a | $1.03 \pm 0.15$ a |
| Solvated methane (millimolar) * | $23.3 \pm 4.49$ a | $25.0 \pm 3.22$ a | $32.6 \pm 4.35$ a | $25.9 \pm 3.34$ a |
| Ammonium (mg $L^{-1}$) * | $174 \pm 20.3$ a | $175 \pm 21.4$ a | $183 \pm 22.6$ a | $194 \pm 24.2$ a |
| Nitrate (mg $L^{-1}$) * | $1.44 \pm 0.40$ a | $1.15 \pm 0.31$ a | $1.93 \pm 0.52$ a | $1.68 \pm 0.53$ a |
| Nitrite (mg $L^{-1}$) * | $1.54 \pm 0.67$ a | $1.26 \pm 0.44$ a | $1.29 \pm 0.54$ a | $0.95 \pm 0.58$ a |
| Phosphate * | $8.39 \pm 2.01$ a | $8.88 \pm 2.29$ a | $10.6 \pm 2.96$ a | $10.3 \pm 2.81$ a |

* Data represent the mean $\pm$ standard error of the mean of 28 weekly determinations. [†] Data represent the mean $\pm$ standard error of the mean of 32 weekly determinations. Within a row, means labelled by the same letter are not significantly different by analysis of variance with Tukey's multiple comparisons test at $p < 0.05$.

## 4. Conclusions

In this experiment, we found that brewers' and distillers' spent grains are suitable substrates for the production of biogas. Seeding of the digesters with animal manure helps to speed digester startup and subsequently ensure stable digester operation. This may be due to the addition of metallic elements that act as required enzymatic cofactors but is also likely due to the addition of methanogenic consortia and ensuring that an adequate amount of bicarbonate buffering is present that resists excessive acidification of digestate. Transmembrane aeration at a pressure of 100 mm Hg significantly increased biogas production as compared to a control treatment, whereas replacing the aeration with $N_2$ at a pressure of 26 mm Hg decreased biogas production as compared to the control. Transmembrane aeration at a pressure of 26 mm Hg also increased biogas production, but not significantly compared to the control system. In contrast to previous research [14,15], however, $HCO_3^-$ buffering was not enhanced by recirculation through the aerated membrane nor was biogas quality enhanced. This was likely due to net gas efflux from the aerated recirculation tanks and retention of solvated $CO_2$ in the digesters. This study showed that BSG and DSG can be used as a viable feedstock to an anaerobic digester given that the pH and bicarbonate buffering are sufficient.

**Author Contributions:** Conceptualization, Z.P.B. and J.H.L.; methodology, Z.P.B., J.H.L. and N.C.L.; formal analysis, Z.P.B. and J.H.L.; validation, J.H.L., S.B., E.D.C., N.C.L. and K.R.S.; investigation, Z.P.B.; data curation, Z.P.B. and J.H.L.; writing—original draft preparation, Z.P.B. and J.H.L.; writing—review and editing, Z.P.B., J.H.L., S.B., E.D.C., N.C.L. and K.R.S.; visualization, Z.P.B. and J.H.L.; supervision, J.H.L.; project administration, J.H.L. funding acquisition, J.H.L. and K.R.S. All authors have read and agreed to the published version of the manuscript.

**Funding:** This research was funded by the U.S. Department of Agriculture, Agricultural Research Service (Project No. 5040-12630-006-00D) and through a USDA-WKU cooperative agreement (Project Number: 5040-12630-006-30-S).

**Institutional Review Board Statement:** Not applicable.

**Informed Consent Statement:** Not applicable.

**Data Availability Statement:** Not applicable.

**Acknowledgments:** The authors thank Stacy Antle and Mike Bryant (USDA-ARS) for technical assistance and Janet Bryant for illustration. Mention of a trademark or product anywhere in this article is to describe experimental procedures, does not constitute a guarantee or warranty of the product by the USDA, and does not imply its approval to the exclusion of other products or vendors that may also be suitable.

**Conflicts of Interest:** The authors declare no conflict of interest.

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
