# Peer review of "Improving Anaerobic Digestion of Brewery and Distillery Spent Grains through Aeration across a Silicone Membrane"

_sustainability, doi:10.3390/su14052755_

Round 1

Reviewer 1 Report

The authors provide a revised version of the first manuscript submitted taking into account all the comments and requests of amendment made in the previous review. The authors provide detailed and justified answers, they provide some amendments and additional information. I agree with all the answers.

Author Response

We thank you for your work on this manuscript.

Sincerely

John Loughrin

Reviewer 2 Report

This study addresses the processing of organic waste from independent breweries and distilleries using low-cost biological processes. These could also be used for the processing of similar type of organic waste. However, please consider the following comments and suggestions for the improvement of  the quality of the manuscript.

Please change the title. A suggestion could be "Improvement of the Anaerobic Digestion of Brewery and Distillery by Recirculation Through a Pressurized Membrane".

Line 21: Please be specific about the type of animal manure that was used for seeding

Line 78: Can you explain how animal manure contributes to enhanced buffering or mineral supplementation throughout the study? This statement should be used carefully?

Line 110: Was the greenhouse's temperature conditioned? If not, reporting a fixed temperature could be misleading.

Line 111: What is the effect of the natural light cycle in anaerobic digestion?

Line 177: Was an assessment of the variation of the organic loading rate performed in this study? If not, why not? 

Figure 2: Please add a legend to quadrants of this figure

Author Response

We thank all the reviewers for their work on this manuscript. We have addressed reviewer 2’s comments as follows:

This study addresses the processing of organic waste from independent breweries and distilleries using low-cost biological processes. These could also be used for the processing of similar type of organic waste. However, please consider the following comments and suggestions for the improvement of  the quality of the manuscript.

Please change the title. A suggestion could be "Improvement of the Anaerobic Digestion of Brewery and Distillery by Recirculation with a Pressurized Membrane". We changed the title to “Improving Anaerobic Digestion of Brewery and Distillery Spent Grains through Aeration Across a Silicone Membrane”

Line 21: Please be specific about the type of animal manure that was used for seeding. We changed this line to: “A combination of dairy and swine manure was used as seeding to provide a methanogenic consortium and bicarbonate buffering.”

Line 78: Can you explain how animal manure contributes to enhanced buffering or mineral supplementation throughout the study? This statement should be used carefully? We amended this sentence to: “Therefore, whether due to enhanced buffering or mineral supplementation, the addition of animal manure to BSG digestate is likely to enhance digester stability during digester startup when buffering is low, and minerals may be lacking due to the relatively intact nature of the grains following alcoholic fermentation.” This change was made to emphasize that the manure was used only to enhance buffering and mineral availability during the startup of the digesters.

Line 110: Was the greenhouse's temperature conditioned? If not, reporting a fixed temperature could be misleading. This is a good point. We changed the sentence to: “The digesters were kept in a greenhouse thermostated at 26.7 °C, with a range of approximately 22.0 to 31.1 °C and using a natural light cycle.

Line 111: What is the effect of the natural light cycle in anaerobic digestion? We don’t know how the how light could affect anaerobic digestion other than by promoting the growth of photosynthetic bacteria. However, we added a sentence in the same paragraph, “The barrels were painted dark green to limit light penetration.”

Line 177: Was an assessment of the variation of the organic loading rate performed in this study? If not, why not? No there wasn’t. We felt this was somewhat of a secondary consideration for this paper as there are many manuscripts that deal in depth with the effect of organic loading rate on anaerobic digestion and particularly because all the digesters were fed at the same rate.

Figure 2: Please add a legend to quadrants of this figure. The y-axis of each quadrant is labelled with the quantity measured, e.g., bicarbonate, pH, etc. with the legend for the graphs put in the figure caption. The graph is somewhat complex, and we feel it is easier to understand as it is laid out now.

Reviewer 3 Report

Dear authors,

I appreciate the revised version of your work, the accuracy of the answer to the suggestions and the improvement of reference section.

Author Response

We thank you for your work on the manuscript.

Sincerely'

John Loughrin

This manuscript is a resubmission of an earlier submission. The following is a list of the peer review reports and author responses from that submission.

Round 1

Reviewer 1 Report

This paper, entitled Optimizing Anaerobic Digestion of Brewery and Distillery Residuals by Recirculation Through a Pressurized Aerated Membrane, is a scholarly work and can increase knowledge on this domain. The authors provide an interesting study, the content is original and relevant to Sustainability. The manuscript is quite well written and well related to existing literature. The abstract and keywords are meaningful.

I have some specific and general comments:

  • Please provide a detailed scheme for digester descriptions and setup
  • Please provide error bars in Figure 1
  • Authors mentioned that "The digesters were kept in a greenhouse maintained at approximately 26.7 °C...", could we consider that anaerobic digestion experiments were carried out under mesophilic conditons? Why choosing this temperature? What about the temperature regulation in a greenhouse?
  • Is there any phase separation between recirculation?
  • How were collected and stored the substrates before experiments?
  • What is the inoculum used for anaerobic digesters?
  • How were determined the loading rate for feeding the anaerobic digesters?
  • In Table 1 and in the text, please provide the methane and biogas production expressed in normal cubic meters or normal liters per gram of VS (1 atm, 273.15 K).
  • Please provide standard deviation of data in Table 3
  • Please provide costs analysis and energy balance of such technical solution
  • Is there any experiments scheduled at pilot scale or industrial scale? What about the ability to transfer this solution at highest scale?

As it this paper is not fully acceptable for publication and requires revision and amendments.

Author Response

I have some specific and general comments:

  • Please provide a detailed scheme for digester descriptions and setup

We provided an illustration of the digesters as Figure 1.

  • Please provide error bars in Figure 1

The only part of Figure 1 we could provide error bars for are for bicarbonate and free, solvated CO2. The pH measurements  are single readings, and the short chain fatty acids are duplicate measurements. But since we are not trying to make weekly comparisons between the four treatments we feel it is best to present the error terms only in Table 1 where statistical tests are presented. In addition, the error bars in sub graphs A & B would make the graphs very hard to read.

Authors mentioned that "The digesters were kept in a greenhouse maintained at approximately 26.7 °C...", could we consider that anaerobic digestion experiments were carried out under mesophilic conditons? Why choosing this temperature? What about the temperature regulation in a greenhouse?

Mesophilic digestion is usually considered to range from 30 to 38 degrees centigrade. Some researchers consider ‘low’ temperature digestion (psychrophilic) to have a range of 0-20 degrees C (e.g. Kashyap et al. Bioresource Technol. 87, 147-153), others to range up to 30 degrees C. We chose this temperature to keep the digesters warm both in the summer and the winter. The environmental controls were able to maintain this temperature in the winter without costing too much money, and able to cool to this temperature in the summer. The greenhouse temperature regulation was quite good with a range of about +/- three degrees. 

  • Is there any phase separation between recirculation?

No

  • How were collected and stored the substrates before experiments?

At the end of the first paragraph of section 2.3, the sentence “All feedstocks were stored at 4 °C prior to use.” was added.

  • What is the inoculum used for anaerobic digesters?

The inoculum was the swine and cow manure.

  • How were determined the loading rate for feeding the anaerobic digesters?

The loading rate was semi-arbitrary but was based partially on the previous feed rate used in Loughrin et al. [2013] and availability of feedstock.

  • In Table 1 and in the text, please provide the methane and biogas production expressed in normal cubic meters or normal liters per gram of VS (1 atm, 273.15 K).

We added paragraphs to section 3.2., Biogas Production and Wastewater Analysis, discussing methane production per gram of VS and a discussion of the limitations of such a measurement for a continuously-fed digester.

  • Please provide standard deviation of data in Table 3

ICP was performed on only two samples of grain or manure and therefore we cannot calculate standard deviations for these samples. The number of determinations for ICP were added in the footnote to Table 3 and to section 2.4

  • Please provide costs analysis and energy balance of such technical solution

We feel that such an analysis is beyond the scope of the study. The recirculation, however, would add very minimal cost to a standard anaerobic digester.

  • Is there any experiments scheduled at pilot scale or industrial scale? What about the ability to transfer this solution at highest scale?

No pilot scale or industrial scale experiments exist to our knowledge. We feel, however, that it would be technically simple and economically viable to perform these experiments at much larger scales.

Reviewer 2 Report

The study is very interesting, but the author can improve the introduction with the explanation of: aim, approach and novelty of the study.

In the section methods the description of the digester is quite complex, try to explain it in a less complex form

Author Response

As it this paper is not fully acceptable for publication and requires revision and amendments.

The study is very interesting, but the author can improve the introduction with the explanation of: aim, approach and novelty of the study.

We modified the introduction a bit to try and emphasize this.

In the section methods the description of the digester is quite complex, try to explain it in a less complex form.

As per our response to reviewers #1 (and #3) we have added a representation of the digester to make the design easier to understand.

Reviewer 3 Report

The topic of the manuscript is important but reviewer couldn’t find novelty of the study. This is very important of scientific research paper. Please describe it more clearly.

Specific comments

  1. Please make one figure of reactor configurations. The system is complicated and difficult to understand only on the text. Especially the recirculation system using silicone hose.
  2. Please describe the operational parameters of the reactor such as HRT, recirculation rate, etc..
  3. Section 3.3 Feedstock ICP : This section describes the concentration of metals in the feedstock. The description is very general things and only Table 3 is enough.
  4. Please discuss the mechanism of transmembrane aeration for increased biogas production more in detail.
  5. Conclusion: “Seeding of the digesters with animal manure helps to speed digester startup and subsequently ensure stable digester operation” Which experiment supports this description?

Author Response

  1. Please make one figure of reactor configurations. The system is complicated and difficult to understand only on the text. Especially the recirculation system using silicone hose.

As per our response to reviewers #1 (and #2) we have added a representation of the digester to make the design easier to understand.

  1. Please describe the operational parameters of the reactor such as HRT, recirculation rate, etc..

HRT =volume of main digester/volume fed week, express as days

Recirculation rate was 100 mL per minute (section 2.2) which resulted in a nominal recirculation or turnover of the digestate once every 1500 minutes (section 2.2). The hydraulic retention time is now given in the last paragraph of section 2.2.

  1. Section 3.3 Feedstock ICP : This section describes the concentration of metals in the feedstock. The description is very general things and only Table 3 is enough.

There is considerable interest in the role of metals in ensuring optimum biogas production in anaerobic digestion. We thought it would be of interest to present the results of ICP analysis in this paper and discuss current knowledge of the role of metals in methane production.

  1. Please discuss the mechanism of transmembrane aeration for increased biogas production more in detail.

It likely is the result of microaeration delivered through the silicone membrane as discussed in the second paragraph of section 3.1.

  1. Conclusion: “Seeding of the digesters with animal manure helps to speed digester startup and subsequently ensure stable digester operation” Which experiment supports this description?

 We added reference [5] to the second to the last paragraph of the Introduction where Goberna et al. found that manure addition did speed up anaerobic digester startup. We should have emphasized this more in the original submission.

Reviewer 4 Report

This study is relevant to the processing of organic waste of independent breweries and distilleries using low-cost biological processes, which could also be adopted for the processing of similar type of waste. However, please consider the following comments and suggestions for the improvement of  the quality of the manuscript. 

Line 47: Why is anaerobic digestion chosen over landfilling, municipal sewerage treatment system, and bio-recycling? Please explain why anaerobic digestion is a better option than the aforementioned alternatives here. Furthermore, BSG could be used as animal feed or turned into a fertilizer without complex processing requirements, which also translate to a low processing cost. Why selecting anaerobic digestion given the fact that most of the BSG nutrient content is lost during the brewing process?

Line 97: There are many ways the stability of anaerobic digestion system could be assessed. One good way is the VFA/Alkalinity ratio. Why wasn't it used in this study?

line 99: Was there any setting to ensure that the temperature inside the digester was maintained at a mesophilic range?

Was the water mixed with feed analyzed?

Line 187:  How often and how many samples were collected for each analysis?

Line 236: How can you ascertain there was loss of CH4?

A very important step is anaerobic digestion is the hydrolysis of the feed. This is a limiting step that often requires the pre-processing of the feed prior to anaerobic digestion. Given the composition of BSG, what processing step was used to ensure fast and conducive hydrolysis?

Please add a legend to figures 1.a, b, c, and d. Additionally, some of the tables (1 to 4) could be turned into a figure for better intuition and comparison between the treatment option

Overall, the novelty of this study and its relevance to the current literature should be further highlighted. Furthermore, a comparison of the results achieved in this study (biogas production given the substrate amount) should be compare against similar studies. 

Author Response

This study is relevant to the processing of organic waste of independent breweries and distilleries using low-cost biological processes, which could also be adopted for the processing of similar type of waste. However, please consider the following comments and suggestions for the improvement of  the quality of the manuscript. 

Line 47: Why is anaerobic digestion chosen over landfilling, municipal sewerage treatment system, and bio-recycling? Please explain why anaerobic digestion is a better option than the aforementioned alternatives here. Furthermore, BSG could be used as animal feed or turned into a fertilizer without complex processing requirements, which also translate to a low processing cost. Why selecting anaerobic digestion given the fact that most of the BSG nutrient content is lost during the brewing process? Actually, we are not really stating that biogas production is better than the other options but that it can be an option that brewers and distillers could consider. Each situation is unique, and space, disposal costs (composting or land filling), availability of livestock, etc. could all play a part in the decision of the most advantageous means for spent grain disposal. We added a line discussing the possibility of using spent grains as livestock feed and how anaerobic digestion might be a means of producing energy to defray some of the costs of disposal.

Line 97: There are many ways the stability of anaerobic digestion system could be assessed. One good way is the VFA/Alkalinity ratio. Why wasn't it used in this study?

Alkalinity is complex to accurately measure but an estimate is usually obtained by titration of the sample with a weak or strong acid depending on the anticipated buffering capacity of the sample and the results are reported as HCO3- and Na2CO3 equivalents (Standard methods for the examination of water and wastewater - APHA, WEF 1998). The results obtained by this titration can be influenced by other buffering species such as ammonia and phosphates. Although both alkalinity and VFA can be estimated by titration (Ciotola et al Energies 2014, 7) we estimate bicarbonate by a gas chromatographic method (Loughrn et al., Front. Environ. Sci, 2017, 5) and VFA by HPLC. Since we measure VFA and alkalinity (as HCO3-) by different methods we get different answers that do other researchers. Average VFA/ALK by our methods yielded ratios of 0.53, 1.65, 2.25, and 0.95 for the no recirculation, N2-recirculation, 26-mm air, and 100-mm air treatments, respectively. Using the titration method, a ratio that is much above 0.4 is considered to be unstable. If VFA concentrations rise too high, therefore, bicarbonate and pH will fall. We feel that our method is adequate for judging digester stability.

line 99: Was there any setting to ensure that the temperature inside the digester was maintained at a mesophilic range? Mesophillic anaerobic digestion is maintained at 20 to 40 degrees centigrade. We maintained the greenhouse at approximately 26.7 degrees C but we are sure the temperature was a bit higher during the day than at night due to solar heating. Other than the greenhouse being maintained at a minimum temperature of 26.7 degrees C and cooling being set to at 29.4 degrees, no other measures were made to ensure digester temperature. The cooling temperature has been added at line 105.

Was the water mixed with feed analyzed?  No. Measurements of volatile solids content were made of the feedstock, however.

Line 187:  How often and how many samples were collected for each analysis?

The dissolved gases and headspace gases were triplicate analyses. This has been added to line 205. VFA and inorganic ions were determined on duplicate samples collected once weekly. This has been made clear on lines 218-228. We neglected to add the details of ion chromatography before and have now added this on lines 224-228.

Line 236: How can you ascertain there was loss of CH4?  Starting on line 291 we state there was little to no loss of methane or carbon dioxide and this assertion is justified by measurement of these gases in their dissolved state in the recirculation tank.

A very important step is anaerobic digestion is the hydrolysis of the feed. This is a limiting step that often requires the pre-processing of the feed prior to anaerobic digestion. Given the composition of BSG, what processing step was used to ensure fast and conducive hydrolysis? There was no processing of the spent grains in this study, however, all 4 treatments were given the same feed so there was no bias introduced.

Please add a legend to figures 1.a, b, c, and d. Additionally, some of the tables (1 to 4) could be turned into a figure for better intuition and comparison between the treatment option. Figure 2 in the present form of the manuscript is labelled A, B, C, D with an explanation in the caption. For clarity’s sake, the Y-axis in each of these shows the quantity being measured. In the case of the ICP, and VFA measurements the individual species being measured vary in concentrations so much that a figure would be more difficult to interpret than a table.

Overall, the novelty of this study and its relevance to the current literature should be further highlighted. Furthermore, a comparison of the results achieved in this study (biogas production given the substrate amount) should be compare against similar studies. On lines 351-375 we do a comparison of gas yields to other studies in terms of biochemical methane yield.

Round 2

Reviewer 1 Report

The authors provide a revised version of their manuscript taking into account all the comments and requests of amendments. I agree with all the answers provided and the manuscript is now fully acceptable for publication. I recommend the following decision: ACCEPT.

Author Response

We thank the reviewer for their work on this manuscript.

Reviewer 3 Report

Still, I couldn’t find the novelty of the study. This is very important for scientific research. Please describe it more clearly.

There is still lacking the discussion for the detailed mechanism of transmembrane aeration for increased biogas production. Section 3.1 is mostly described tube properties.

Author Response

Comments and Suggestions for Authors

Still, I couldn’t find the novelty of the study. This is very important for scientific research. Please describe it more clearly.

In lines 77-96, an extra paragraph was added emphasizing why recirculation through the silicone hose was done, i.e., to see if biogas quality could be enhanced through boosting bicarbonate buffering. This was also stressed further in the last paragraph of the introduction. Also, in the conclusions sections the line “In contrast to previous research [14,15], however, HCO3- buffering was not enhanced by recirculation through the aerated membrane nor was biogas quality enhanced.” In order to contrast it to previous research.

There is still lacking the discussion for the detailed mechanism of transmembrane aeration for increased biogas production. Section 3.1 is mostly described tube properties.

The properties of the membrane are critical to interpreting results. In lines 286-287 we inserted the statement “As will be seen below, this had consequences on the concentrations of HCO3- in the digesters subjected to aeration through the silicone membrane.” Also, the lines “In contrast to previous research [14,15] HCO3- buffering was not enhanced relative to the control treatment and therefore neither was biogas CO2 decreased. As stated in section 3.1, this is likely due to retention of sCO2 in the digester due to net gas influx from the aerated recirculation tank into the digester. (Lines 328-332.). We also added the line “This was likely due to net gas efflux from the aerated recirculation tanks and retention of solvated CO2 in the digesters.” In the conclusions (Lines 489-490)

Reviewer 4 Report

Thanks for considering the suggestions. I noticed an improvement in the quality of the manuscript. 

Author Response

Starting on line 241, a sentence was added to further clarify the expected results and how the results obtained differed from past results.

On line 305 we added the word aerated to make the discussion clearer.

On line 364,  the sentence “With no O2 diffusion into the N2 treated digester and its  subsequent conversion into HCO3-, buffering and pH remained low.” was added as extra interpretation of the results.

Starting on line  351, a sentence was added to stress the difference in the present results as compared to previous research and the reason for the differences found.

Round 3

Reviewer 3 Report

na

Author Response

na